# Hyperquaternions and physics

**Patrick R. Girard[1★], Romaric Pujol[2], Patrick Clarysse[1] and Philippe Delachartre[1]**

**1** CREATIS, Université de Lyon, CNRS UMR5520, INSERM U1294, INSA-Lyon, France
**2** Pôle de Mathématiques, INSA-Lyon, France

★ patrick.girard@creatis.insa-lyon.fr

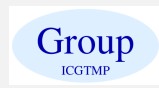
## Abstract

The paper develops, within a new representation of Clifford algebras in terms of tensor products of quaternions called hyperquaternions, several applications. The first application is a quaternion 2D representation in contradistinction to the frequently used 3D one. The second one is a new representation of the conformal group in (1+2) space (signature +−−) within the Dirac algebra $C_5(2,3) \simeq \mathbb{C} \otimes \mathbb{H} \otimes \mathbb{H}$ subalgebra of $\mathbb{H} \otimes \mathbb{H} \otimes \mathbb{H}$. A numerical example and a canonical decomposition into simple planes are given. The third application is a classification of all hyperquaternion algebras into four types, providing the general formulas of the signatures and relating them to the symmetry groups of physics.

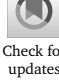

# 1 Introduction

A new hyperquaternionic representation of Clifford algebras [1, 2] has been introduced recently [3–7], hyperquaternion algebras being defined as tensor products of quaternion algebras (or subalgebra thereof). This paper develops several hyperquaternionic applications.

Throughout this paper, $\mathbb{H}^{\otimes m}$ will denote the tensor product of $m$ quaternion algebras, i.e. $\mathbb{H}^{\otimes m} = \mathbb{H} \otimes \mathbb{H} \otimes \cdots \otimes \mathbb{H}$ ($m$ terms). The structure of the paper is as follows.

In the preliminaries, the historical origins and basic concepts of hyperquaternion Clifford algebras are examined. In the third section, a quaternion $2D$ representation is proposed in contradistinction to a widely used $3D$ representation. In the fourth section, the conformal group of the $(1 + 2)$ space (signature $+--$) is developed within the Dirac algebra $C_5(2,3) \simeq \mathbb{C} \otimes \mathbb{H}^{\otimes 2}$ considered as a subalgebra of $\mathbb{H}^{\otimes 3} \simeq C_6(2,4)$. The choice of the $(1 + 2)$ space is motivated by its use in quantum gravity [8]. A numerical example together with a canonical decomposition into simple planes is provided. Finally, the fifth section gives a classification of all hyperquaternion algebras into four types with general formulas of the signatures and associated symmetry groups of physics.

# 2 Preliminaries: Clifford algebras and hyperquaternions

## 2.1 Quaternions and biquaternions

The quaternion group [9, 10] was discovered in 1843 by W. R. Hamilton and is constituted by the elements $(\pm 1, \pm i, \pm j, \pm k)$ satisfying the formula

$$i^2 = j^2 = k^2 = ijk = -1. \tag{1}$$

The quaternion algebra $\mathbb{H}$ is defined as a set of four real numbers $q_i$, called quaternions $q = q_0 + q_1 i + q_2 j + q_3 k$. The conjugate $q_c$ of $q$ is defined by $q_c = q_0 - q_1 i - q_2 j - q_3 k$. Hamilton was to give a $3D$ (if not $4D$) interpretation of quaternions which was to lead to the classical vector calculus still in use today. He also introduced complex quaternions which he named biquaternions.

## 2.2 Clifford algebras and hyperquaternions

Clifford in 1878, introduced his algebras as tensor products of quaternion algebras [11]. He proved the following theorem

$$C_{2m} \simeq \mathbb{H}^{\otimes m}, \quad C_{2m-1} \simeq \mathbb{C} \otimes \mathbb{H}^{\otimes m-1}. \tag{2}$$

Lipschitz in 1880, derived the rotation formula of $nD$ Euclidean spaces [12]

$$x' = axa^{-1}, \quad a \in C^+. \tag{3}$$

He thereby rediscovered the (even) Clifford algebras. In 1922, Moore [13] was to call Lipschitz's algebras: hyperquaternions, a term which we shall extend to all Clifford algebras. A major success of Clifford algebras in physics was the Dirac algebra and the spinor calculus. Recent developments in Clifford algebras seem to have somewhat neglected if not totally ignored the hyperquaternionic filiation.

In terms of generators, the Clifford algebra $C_n(p,q)$ has $n = p + q$ generators $e_i$ such that $e_i e_j + e_j e_i = 0$ ($i \neq j$), $e_i^2 = +1$ ($p$ generators) and $e_i^2 = -1$ ($q$ generators). The total number of elements is $2^n$. The algebra contains scalars ($S$), vectors ($V$) $e_i$, bivectors ($B$) $e_i e_j$ ($i \neq j$), etc.

$C^+$ is the (even) subalgebra constituted by products of an even number of $e_i$. It is to be noticed that the hyperquaternion product is independent of the choice of the generators whereas the multivector structure depends on it.

Examples of hyperquaternion Clifford algebras are: quaternions $\mathbb{H}$ ($e_1 = i, e_2 = j$), biquaternions $\mathbb{C} \otimes \mathbb{H}$ ($e_1 = iI, e_2 = jI, e_3 = kI, I = 1 \otimes i$), $\mathbb{H} \otimes \mathbb{H}$ ($e_0 = j, e_1 = kI, e_2 = kJ, e_3 = kK$) with $(I, J, K) = 1 \otimes (i, j, k)$.

## 3 Quaternion 2D representation

In contradistinction to Hamilton who gave a $3D$ interpretation of quaternions which is still widely used today, we shall provide a $2D$ plane representation below since quaternions constitute a Clifford algebra with only two generators

$$e_1 = i, \quad e_2 = j, \quad e_1 e_2 = k \quad \left(e_1^2 = e_2^2 = -1\right). \tag{4}$$

Interior and exterior products can be defined with $x = x_1 i + x_2 j \in V, B = bk$ bivector ($b \in \mathbb{R}$) by

$$x.y = -(xy + yx)/2 = x_1 y_1 + x_2 y_2 \in S, \tag{5}$$
$$x \wedge y = (xy - yx)/2 = (x_1 y_2 - x_2 y_1)k \in B, \tag{6}$$
$$x.B = -(xB - Bx)/2 = b(-x_2 i + x_1 j) \in V, \tag{7}$$
$$x \wedge B = (xB + Bx)/2 = 0. \tag{8}$$

The rotation group $SO(2)$ is expressed by

$$x' = rxr_c = (x_1 \cos\theta - x_2 \sin\theta)i + (x_1 \sin\theta + x_2 \cos\theta)j, \tag{9}$$

with

$$r = e^{k\theta/2} = (\cos\theta/2 + k\sin\theta/2) \in B. \tag{10}$$

The modeling of an Euclidean $3D$ space can be realized similarly with biquaternions [14].

## 4 Hyperquaternionic conformal group in (1+2) space

The conformal group of the $(1 + 3)$ space has been examined within the algebra $\mathbb{H}^{\otimes 3} \simeq C_6(2, 4)$ in [4]. Here, we consider the $(1 + 2)$ subspace within the subalgebra $C_5(2, 3) \simeq \mathbb{C} \otimes \mathbb{H}^{\otimes 2} \simeq \mathbb{C}(4)$ isomorphic to the Dirac algebra. This space has received much attention in particular with respect to quantum gravity [8]. We first introduce the algebraic structure, then the restricted conformal group and a numerical example including a canonical decomposition into simple planes.

### 4.1 Algebraic structure

As generators of the subalgebra $C_5(2, 3) \simeq \mathbb{C} \otimes \mathbb{H}^{\otimes 2}$, we take

$$e_a = kI, \quad e_0 = kJ, \quad e_1 = kKl, \quad e_2 = kKm, \quad e_b = j, \tag{11}$$

with

$$\mathbb{H}^{\otimes 3} = (i, j, k) \otimes (I, J, K) \otimes (l, m, n), \tag{12}$$

and $(l, m, n) = 1 \otimes 1 \otimes (i, j, k)$. A general element $A$ of $\mathbb{H}^{\otimes 3}$ can be viewed as a set of 16 quaternions $[q_i] = a_i + b_i l + c_i m + d_i n$

$$
\begin{aligned}
A = {}& [q_1] + I[q_2] + J[q_3] + K[q_4] + i[q_5] + iI[q_6] + iJ[q_7] + iK[q_8] \\
& + j[q_9] + jI[q_{10}] + jJ[q_{11}] + jK[q_{12}] + k[q_{13}] + kI[q_{14}] + kJ[q_{15}] + kK[q_{16}] .
\end{aligned} \tag{13}
$$

The explicit multivector structure of $C_5(2, 3)$ is given in Appendix [A]. The algebra has $2^5 = 32$ elements with 10 parameters for the bivectors. The product is implemented in http://www.notebookarchive.org/2021-08-6z1zbda/.

## 4.2 Restricted conformal group

The restricted conformal group in $(1 + 2)$ space is obtained via the procedure described in [1].

First, one constructs an affine space within $C_5(2, 3)$. Let $X$ be a five dimensional vector

$$
X = \frac{(x^2 - 1)}{2} e_a + x + \frac{(x^2 + 1)}{2} e_b = x^2 \varepsilon_1 + x + \varepsilon_2 , \tag{14}
$$

with $x = x_0 e_0 + x_1 e_1 + x_2 e_2 \in E_3, \quad X^2 = 0$ and

$$
\varepsilon_1 = \frac{e_a + e_b}{2} , \quad \varepsilon_2 = \frac{e_b - e_a}{2} , \quad \varepsilon_1^2 = \varepsilon_2^2 = 0 . \tag{15}
$$

The restricted conformal group is then expressed by the transformations

$$
X' = a X a_c \quad (a a_c = 1, \quad a \in C_5^+(2, 3)) . \tag{16}
$$

They are composed of

- spatial rotations $a = e^{n \frac{\theta}{2}}$ ,

- boosts $a = e^{B \frac{\theta}{2}}, \quad B \in (Il, Im)$ ,

- translations $a = e^{\varepsilon_1 u} = 1 + \varepsilon_1 u \quad (u \in E_3)$ ,

- transversions $a = e^{\varepsilon_2 v} = 1 + \varepsilon_2 v \quad (v \in E_3)$ ,

- dilations $a = e^{e_a e_b \frac{\varphi}{2}} = e^{-iI \frac{\varphi}{2}} = \cosh \frac{\varphi}{2} - iI \sinh \frac{\varphi}{2}$ .

The total number of parameters is $\frac{(n+2)(n+1)}{2} = 10 \ (n = 3)$ . Through combinations, one obtains the general transformations

$$
X' = f X f_c \quad (f f_c = 1, \quad f \in C_5^+(2, 3)) . \tag{17}
$$

The Lie algebra is given in [4]

## 4.3 Numerical example

Here, we present a numerical example consisting of a set of transformations together with a canonical decomposition thereof.

As transformation $X' = f X f_c$ we shall consider a dilation ($e^{-\varphi} = 1/3$) followed by a unit translation ($u = e_1$) and a rotation ($\theta = \pi/2$ in the plane $e_{12} = n$). The combination of these transforms yields the hyperquaternion $f \in C^+$

$$f = e^{n\frac{\theta}{2}} e^{\varepsilon_1 u} e^{-iI\frac{\varphi}{2}} \tag{18}$$

$$= (\cos\theta/2 + n\sin\theta/2)(1 + \varepsilon_1 u)(\cosh\varphi/2 - iI\sinh\varphi/2) \tag{19}$$

$$= \left(\frac{1}{\sqrt{2}} + n\frac{1}{\sqrt{2}}\right)\left[1 + \frac{(kI+j)}{2}kKl\right]\left(\frac{2}{\sqrt{3}} - iI\frac{1}{\sqrt{3}}\right) \tag{20}$$

$$= \left(\sqrt{\frac{2}{3}} - \frac{1}{\sqrt{6}}iI\right)(1+n) + \frac{1}{2}\sqrt{\frac{2}{3}}(J+iK)(l+m), \tag{21}$$

with $\tan\frac{\theta}{2} = 1$, $\tanh\frac{\varphi}{2} = \frac{1}{2}$ $\left(e^{\varphi} = \frac{1+th\varphi/2}{1-th\varphi/2} = \frac{3/2}{1/2} = 3\right)$.

The bivector part $B$ of $f$ generating the transformation, divided by the scalar $\sqrt{\frac{2}{3}}$ is

$$B = n - \frac{iJ}{2} + \frac{3}{4}(J+iK)(l+m). \tag{22}$$

The canonical decomposition [4] of $B$ and $f$ into simple, orthogonal and commuting planes $(B_1, B_2)$ with $b_1 = \tan\frac{\Phi_1}{2} = 1$, $b_2 = \tanh\frac{\Phi_2}{2} = \frac{1}{2}$ leads to

$$B = b_1 B_1 + b_2 B_2, \quad f = e^{\frac{\Phi_1}{2}B_1} e^{\frac{\Phi_2}{2}B_2}, \tag{23}$$

with

$$B_1 = n + \frac{3}{10}(J+iK)(3l+m), \quad B_1^2 = -1, \tag{24}$$

$$B_2 = \frac{3}{10}(J+iK)(-l+3m) - IK, \quad B_2^2 = 1. \tag{25}$$

The two invariants of the transformation are

$$S_1 = B.B = -\frac{3}{4}, \quad S_2 = [(B \wedge B).B].B = -1. \tag{26}$$

The conformal transformation with $X = e_a + e_1 + e_b$ ($x_0 = 0, x_1 = 1, x_2 = 0$) is obtained either directly

$$e_1 \xrightarrow{D} e_1/3 \xrightarrow{T} (1/3 + 1)e_1 = (4/3)e_1 \xrightarrow{R} (4/3)e_2, \tag{27}$$

or by computation:

$$X' = f X f_c = x'_a e_a + x' + x'_b e_b \tag{28}$$

$$= -\frac{25}{6}e_a + 4e_2 - \frac{7}{6}e_b, \tag{29}$$

yielding the final transform

$$x \to y(x) = \frac{x'}{x'_b - x'_a} = \frac{4}{3}e_2. \tag{30}$$

## 5 Classification of hyperquaternion algebras

Table 1 lists a few hyperquaternion algebras and their signature $(p, q)$ obtained via the generators given in [7]. The table shows the importance of the parameter $s = p - q$ [2, 15]. It

reveals four classes of hyperquaternions: the algebras $\mathbb{H}^{\otimes r}$ ($r$ even or odd) and the subalgebras $C^+$. From $(n,s)$ one deduces $p = (n+s)/2, q = (n-s)/2$ yielding the general formulas for $m$ integer $(m \geqslant 1)$

$$\mathbb{H}^{\otimes 2m} \simeq C_{4m}(2m+1, 2m-1), \quad (s=2),$$

$$\mathbb{C} \otimes \mathbb{H}^{\otimes(2m-1)} \simeq C_{4m-1}(2m+1, 2m-2), \quad (s=3),$$

$$\mathbb{H}^{\otimes(2m-1)} \simeq C_{4m-2}(2m-2, 2m), \quad (s=-2),$$

$$\mathbb{C} \otimes \mathbb{H}^{\otimes(2m-2)} \simeq C_{4m-3}(2m-2, 2m-1), \quad (s=-1).$$

All signatures of hyperquaternion algebras can be derived from the first four ones via the formula

$$C_{n+4}(p+2, q+2) = C_n(p,q) \otimes \mathbb{H}^{\otimes 2},$$

resulting from the double application of the general formula

$$C_{n+2}(p+1, q+1) = C_n(p,q) \otimes C_2(1,1),$$

together with $C_2(1,1) \simeq \mathbb{R}(2), \mathbb{R}(2) \otimes \mathbb{R}(2) \simeq \mathbb{R}(4) \simeq \mathbb{H}^{\otimes 2}$. Furthermore, since

$$\mathbb{H}^{\otimes 2} \simeq \mathbb{R}(4), \quad \mathbb{C} \otimes \mathbb{H}^{\otimes 2} \simeq \mathbb{C}(4), \quad \mathbb{H}^{\otimes 3} \simeq \mathbb{H}(4),$$

one obtains all square real, complex and quaternionic matrices. Concerning the matrix representation of hyperquaternion algebras, which is beyond the scope of this paper, the above isomorphisms show that $\mathbb{H}^{\otimes 2}$ can be represented either by a reducible real matrix $\mathbb{R}(16)$ (real $16 \times 16$ matrix) or by an irreducible $\mathbb{R}(4)$ matrix ($\mathbb{H}$ being represented by an irreducible $\mathbb{R}(4)$ matrix). Similarly, $\mathbb{H}^{\otimes 3}$ and its subalgebra $\mathbb{C} \otimes \mathbb{H}^{\otimes 2}$ can be represented either by a reducible matrix $\mathbb{R}(64)$ or by an irreducible matrix $\mathbb{R}(16)$. A classification of real irreducible representations of quaternionic Clifford algebras can be found in [16,17].

A hyperconjugation defined as

$$A_H = (i_c, j_c, k_c) \otimes (I_c, J_c, K_c) \otimes (l_c, m_c, n_c),$$

yields the matrix transposition, adjunction and transpose quaternion conjugate. Finally, writing $\omega = e_1 \cdots e_n$, one obtains for all hyperquaternion algebras with the above values of $s$ and

Table 1: Hyperquaternion algebras (SR: special relativity, RQM: relativistic quantum mechanics, usp: unitary symplectic physics, sm: standard model).

| $C_n(p,q)$ | $n$ | $p$ | $q$ | $s = p-q$ | Group | Physics |
|---|---|---|---|---|---|---|
| $\mathbb{C}$ | 1 | 0 | 1 | $-1$ | $U(1)$ | 1D |
| $\mathbb{H}$ | 2 | 0 | 2 | $-2$ | $USp(1)$ | 2D |
| $\mathbb{C} \otimes \mathbb{H}$ | 3 | 3 | 0 | 3 | $SU(2)$ | 3D |
| $\mathbb{H}^{\otimes 2} \simeq \mathbb{R}(4)$ | 4 | 3 | 1 | 2 | $SO(3,1)$ | SR |
| $\mathbb{C} \otimes \mathbb{H}^{\otimes 2} \simeq \mathbb{C}(4)$ | 5 | 2 | 3 | $-1$ | $SU(4)$ | RQM |
| $\mathbb{H}^{\otimes 3} \simeq \mathbb{H}(4)$ | 6 | 2 | 4 | $-2$ | $USp(4)$ | usp |
| $\mathbb{C} \otimes \mathbb{H}^{\otimes 3}$ | 7 | 5 | 2 | 3 | $SU(8)$ | sm |
| $\mathbb{H}^{\otimes 4}$ | 8 | 5 | 3 | 2 | $SO(5,3)$ | |
| $\mathbb{C} \otimes \mathbb{H}^{\otimes 4}$ | 9 | 4 | 5 | $-1$ | $SU(16)$ | |
| $\mathbb{H}^{\otimes 5}$ | 10 | 4 | 6 | $-2$ | $USp(16)$ | |
| $\mathbb{C} \otimes \mathbb{H}^{\otimes 5}$ | 11 | 7 | 4 | 3 | $SU(32)$ | |
| $\mathbb{H}^{\otimes 6}$ | 12 | 7 | 5 | 2 | $SO(7,5)$ | |

the classical derivation (developing $n = p + q$ and using $(-1)^{pq} = (-1)^{-pq}$)

$$\omega^2 = (-1)^{\frac{n(n-1)}{2}} e_1^2 ... e_n^2 = (-1)^{\frac{n(n-1)}{2}+q} = (-1)^{\frac{s(s-1)}{2}} = -1 \, .$$

Though hyperquaternion algebras have been neglected in the past, recent algebraic software like `Mathematica` and numerical computing have opened perspectives for the hyperquaternion calculus. An advantage of the hyperquaternion representation of Clifford algebras, is that the product is defined independently of the choice of the generators. Furthermore, hyperquaternion algebras single out specific Clifford algebras which seem to be closely related to symmetry group of physics as indicated in the table above. Thus they might constitute a step towards a greater unification as proposed in [18].

# 6 Conclusion

The paper has developed applications of a new hyperquaternionic representation of Clifford algebras in terms of tensor products of quaternion algebras. One advantage of hyperquaternion algebras is a uniquely defined product, independent of the choice of generators. Though, hyperquaternions have been somewhat neglected so far, they have become more accessible due to the introduction of algebraic and numerical computing. As applications, the paper has examined the quaternion $2D$ representation, the conformal group in $(1+2)$ space together with a numerical example and implementation. Finally, a classification of all hyperquaternion algebras into four types has been given, with general formulas of the signatures and the associated symmetry groups. We hope to have shown that the hyperquaternion algebras might constitute a useful unifying tool for physics.

# Acknowledgements

The paper is dedicated to the memory of our colleague and friend Prof. Dr. Robert Goutte (1932-2022) who greatly furthered and collaborated on research on quaternions and hyperquaternions.

The authors acknowledge and thank an anonymous referee for comments clarifying the matrix quaternionic Clifford algebra representation issue.

**Funding information** This work was supported by the LABEX PRIMES (ANR-11-LABX-0063) of Université de Lyon, within the program "Investissements d'Avenir" (ANR-11-IDEX-0007) operated by the French National Research Agency (ANR).

# A    Multivector structure of $C_5(2,3)$

($e_{012} = e_0 e_1 e_2$, etc..)

$$
\begin{bmatrix}
1 & 0 & 0 & n = e_{12} \\
0 & I\ l = e_{10} & I\ m = e_{20} & 0 \\
0 & J\ l = e_{a1} & J\ m = e_{a2} & 0 \\
K = e_{0a} & 0 & 0 & Kn = e_{0a12} \\
0 & il = e_{0a1b} & im = e_{0a2b} & 0 \\
iI = e_{ba} & 0 & 0 & iI\ n = e_{1a2b} \\
iJ = e_{b0} & 0 & 0 & iJ\ n = e_{102b} \\
0 & iKl = e_{b1} & iKm = e_{b2} & 0 \\
j = e_b & 0 & 0 & jn = e_{12b} \\
0 & jI\ l = e_{10b} & jI\ m = e_{b20} & 0 \\
0 & jJ\ l = e_{a1b} & jJ\ m = e_{a2b} & 0 \\
jK = e_{0ab} & 0 & 0 & jKn = e_{0a12b} \\
0 & kl = e_{a01} & km = e_{a02} & 0 \\
kI = e_a & 0 & 0 & kI\ n = e_{a12} \\
kJ = e_0 & 0 & 0 & kJ\ n = e_{012} \\
0 & kKl = e_1 & kKm = e_2 & 0
\end{bmatrix}
$$

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
