# Peer review of "Hyperquaternions and Physics"

_SciPost Physics Proceedings, doi:SciPost Phys. Proc. 14, 030 (2023)_

## Round 1 · Referee Report · Anonymous (Referee 1) · 2022-12-20

Strengths
Weaknesses
Report
The paper is a clear assessment of the properties of hyperquaternions with applications to 2D physics and conformal algebras. The papers could be published, but before publications some points should be addressed by the authors.
1 - In the Introduction, line 9, there is an obvious mistake: the signature of the (1+2) space is +—-, instead of +-.
2 - The hyperquaternionic algebras C_5(2,3) and C_6(2,4) are non-minimal representations of the corresponding Clifford algebras. This point needs to be clarified.
In: - S. Okubo, Real representations of finite Clifford algebras,1. Classification, J. Math. Phys. 32 (1991) 1657., the quaternionic Clifford algebras associated to the different space-time signatures were presented.
In: - H.L. Carrion et al.,Quaternionic and octonionic spinors. A classification, JHEP04(2003)040; arXiv:hep-th/0302113 a table was presented (Table 4 in JHEP, Table 7 in the arXiv version) with the dimension of the quaternionic Clifford algebras.
In real counting (as matrices with real entries) both C_5(2,3) and C_6(2,4) are given by 16x16 matrices.
The hyperquaternionic construction of C_5(2,3) and C_6(2,4) is non-minimal. C_5(2,3) is presented as tensor products of CxHxH which, in real counting, gives matrices with 32x32 real entries.
C_6(2,4) is presented as tensor products HxHxH which, in real counting, gives matrices with 64x64 real entries.
The hyperquaternionic representations of C_5(2,3) and C_6(2,4) are reducible, while the quaternionic representations of C_5(2,3) and C_6(2,4) are known to be irreducible.
It is unclear which advantages would be produced by a reducible hyperquaternionic representation with respect to the reducible quaternionic representation.
Requested changes
Correction of signature at line 9 of the Introduction.
Explanation of the reducibility of the hyperquaternionic reps of the Clifford algebras
C_5(2,3) and C_6(2,4).

---

## Round 2 · List of Changes

1 - In the Introduction, line 9, there is an obvious mistake: the signature of
the (1+2) space is +—-, instead of +-.
This has been corrected.
the (1+2) space is +—-, instead of +-.
This has been corrected.

---

## Round 3 · Referee Report · Anonymous (Referee 1) · 2023-1-9

Report

The authors answered the question raised in the previous report and clarified the relation between quaternionic and hyperquaternionic representations of Clifford algebras.
The paper can be published as is.

---

## Round 3 · List of Changes

1) Introduction (+-) changed into (+--)

2) Section 5 addition of « Concerning the matrix representation of hyperquaternion algebras, which is beyond the scope of this paper, the above isomorphisms show that HH can be represented either by a reducible real matrix R(16) (real 16×16 matrix) or by an irreducible R(4) matrix (H being represented by an irreducible R(4) matrix). Similarly, HHH and its subalgebra CH*H can be represented either by a reducible matrix R(64) or by an irreducible matrix R(16) . A classification of real irreducible representations of quaternionic Clifford algebras can be found in [16,17]. »

Addition of two references [16] S. Okubo, Real representations of finite Clifford algebras. I. Classification, J. Math. Phys. 32, 1657 (1991), doi:10.1088/1126-6708/2003/04/040 [17] H.L. Carrion, M. Rojas, F. Toppan, Quaternionic and Octonionic Spinors. A Classification, JHEP04. 2003, (2003), doi:10.1088/1126-6708/2003/04/040

3) Acknowledgements : addition of « The authors acknowledge and thank an anonymous referee for comments clarifying the matrix quaternionic Clifford algebra representation issue. »

---

## Editorial Decision

published